

# Multi-omics reveals changed energy metabolism of liver and muscle by caffeine after mice swimming

Yang Han[1,2,3,*], Qian Jia[1,3,*], Yu Tian[1], Yan Yan[1], Kunlun He[2,3] and Xiaojing Zhao[1,2,3]

[1] Translational Medical Research Center, Medical Innovation Research Division of Chinese PLA General Hospital, Beijing, China
[2] Key Laboratory of Biomedical Engineering and Translational Medicine, Ministry of Industry and Information Technology, Medical Innovation Research Division of Chinese PLA General Hospital, Beijing, China
[3] Beijing Key Laboratory of Precision Medicine for Chronic Heart Failure, Medical Innovation Research Division of Chinese PLA General Hospital, Beijing, China
[*] These authors contributed equally to this work.

## ABSTRACT

In recent years, numerous studies have investigated the effects of caffeine on exercise, and provide convincing evidence for its ergogenic effects on exercise performance. However, the precise mechanisms underlying these ergogenic effects remain unclear. In this study, an exercise swimming model was conducted to investigate the effects of orally administered with caffeine before swimming on the alterations of proteome and energy metabolome of liver and muscle after swimming. We found proteins in liver, such as S100a8, S100a9, Gabpa, Igfbp1 and Sdc4, were significantly up-regulated, while Rbp4 and Tf decreased after swimming were further down-regulated in caffeine group. The glycolysis and pentose phosphate pathways in liver and muscle were both significantly down-regulated in caffeine group. The pyruvate carboxylase and amino acid levels in liver, including cysteine, serine and tyrosine, were markedly up-regulated in caffeine group, exhibiting a strong correlation with the increased pyruvic acid and oxaloacetate levels in muscle. Moreover, caffeine significantly decreased the lactate levels in both liver and muscle after swimming, potentially benefiting exercise performance.

## INTRODUCTION

Caffeine is a highly popular ergogenic aid (*Aguilar-Navarro et al., 2019*; *Del Coso, Muñoz & Muñoz Guerra, 2011*). Mechanistically, caffeine can reduce fatigue, reaction time and increase alertness by binding to adenosine receptors to antagonize the effects of adenosine (*Clark & Landolt, 2017*). In recent years, as a result of its ergogenic function, numerous studies have explored the effects of caffeine on exercise performance (*Grgic et al., 2020*). Studies have demonstrated caffeine was definitively beneficial to exercise performance (*McLellan, Caldwell & Lieberman, 2016*), such as muscular strength and endurance,

Corresponding authors
Kunlun He, kunlunhe@plagh.org
Xiaojing Zhao, xjingzhao@126.com

and resistance exercise. However, the mechanisms that caffeine can enhance exercise performance are still unclear.

Some studies indicated that caffeine improved performance owing to its local effects directly on skeletal muscle (*Tallis, Duncan & James, 2015*). For another, muscle force production was potentiated directly by caffeine through binding to muscle ryanodine receptor 1 (*Grgic, 2022*). Additionally, caffeine can accelerate contraction-induced metabolic activation, thereby contributing to muscle endurance performance (*Tsuda, Hayashi & Egawa, 2019*). However, these results still could not explain its ergogenic effects on exercise performance accurately and completely. Previous study reported that caffeine could significantly change various metabolic pathways in contracting muscles like pentose phosphate, nucleotide synthesis, $\beta$-oxidation, tricarboxylic acid cycle and amino acid metabolism (*Tsuda, Hayashi & Egawa, 2019*). Notably, these pathways play important roles in energy metabolism. However, a comprehensive energy metabolism profile after caffeine intake has not been reported currently. In addition, caffeine is mainly metabolized by cytochrome P-450 (CYP) enzyme in the liver. Therefore, the effect of caffeine on metabolic pathways related to energy metabolism of liver is necessary to further explore. Given the Cori cycle (also known as the lactic acid cycle), which involves the transport of lactate produced by muscle after swimming to liver, where it is converted back to glucose and returns to muscle as new energy sources, the energy metabolism in muscle is also warranted to further study.

At present, a lot of exercise-related proteins have been identified. Some proteins, such as triosephosphate isomerase, troponin T, and muscle creatine kinase, differentially expressed after acute exercise, are also regulated in exercise-trained muscle (*Burniston & Hoffman, 2011*). Based on the proteomic analysis, a distinct proteomic profile between muscle fiber subtypes has been identified and it may represent the different molecular response under the different physiology conditions after swimming (*Murgia et al., 2015*). Furthermore, study has revealed proteome allocation influenced metabolism during high intensity exercise (*Nilsson et al., 2019*). Consequently, examining the impact of caffeine on the proteome profile post-swimming is warranted, as it may influence the energy metabolome profile.

Herein we explored the effects of caffeine on the energy metabolism and proteome of liver and gastrocnemius muscle after mice swimming. Associations between these outcomes were analyzed to explore the relationship between the liver and muscle after swimming.

## MATERIALS AND METHODS

### Animal experiment and sample collection

The animal experiments were conducted in the Medical Laboratory Animal Center at the Chines PLA General Hospital, which was approved by the Animal Care and Use Committee of Chines PLA General Hospital (SQ2020030). Eighteen 6-week-old male C57BL/6J mice purchased from laboratory animal research center of our hospital were housed under specific pathogen-free condition. Data were collected as previously described in *Han et al. (2022)*. Specifically, the housed temperature and humidity were controlled at 25 $\pm$ 2 °C and 50 $\pm$ 10%, respectively, and a 12:12 h light-dark cycle. The animals were fed a

standard pellet diet and distilled water *ad libitum*. Each mouse was housed in a 30 cm high cage and the floor of the cage covered with an adequate depth of dust free Aspen wood chip. If mice are obviously unwell, the experiment should be terminated. We determined whether animals were unwell by observing behaviors such as silence, lethargy, inactivity, loss of appetite, and refusal to eat. Animals were not be euthanized until the end of the experiment. After the experiment, all animals were euthanized by intraperitoneal injection of pentobarbital sodium at a dose of 200 mg/kg.

Spontaneous wheel running exercise is not suitable for studies that require precise timing for exploring acute exercise adaptations, since the investigator cannot easily regulate the duration or intensity of the running behavior (*Seo et al., 2014*). What's more, it has been proved that caffeine ingestion improves exercise performance in a broad range of exercise tasks (*Grgic et al., 2020*), and we focused on the ergogenic effects of caffeine on acute exercise performance. Therefore, the forced swim test was selected for our study. Animals were randomly divided into the discovery group ($n = 18$) and the validation group ($n = 6$). The discovery group includes the control group ($n = 9$) and the caffeine group ($n = 9$). Mice in the caffeine group were orally administered with one mL of 0.125 mg/mL caffeine solution before mice swimming. The control group was given an equal volume of normal saline. Both the control and caffeine groups of mice were placed in a pool with a temperature of $32 \pm 2\,°C$ and a depth of over 40 cm, swimming for 30 min, ensuring constant movement throughout the procedure. After the first swimming experiment, mice rested for 4 hours before undergoing the same administration and swimming experiment. The validation group was also divided into the control (C_V, $n = 3$) and the caffeine groups (F_V, $n = 3$). The experiment of validation group was conducted independently, following the same procedure as that of the discovery group. Stress-related immobility in the mouse-forced swim test is labeled as a depression-like phenotype (*de Kloet & Molendijk, 2016*). To separate the effects of exercise from stress, we stirred near the mice with a stick to keep them in continuous motion while they were floating on the water with immobilized limbs.

In order to mitigate and abolish the pain and stress experienced by mice during blood collection, and to facilitate the experimental process while ensuring animal safety, after the second swimming experiment, the mice were intraperitoneally injected with a mixture of ketamine and xylazine to achieve sedative and analgesic effects. Once the mouse lost consciousness, blood was extracted from the orbit. To obtain serum, after standing for 40 min without adding anticoagulant, the blood was centrifuged at $4\,°C$ and $1,900 \times g$ for 10 min, and the supernatant was taken and stored at $-80\,°C$. Subsequently, the mice were dissected to obtain liver samples immediately including mice with (the discovery group: FL, the validation group: FL_V) or without (the discovery group: CL, the validation group: CL_V) caffeine, as well as gastrocnemius muscles including mice with (the discovery group: FM) or without (the discovery group: CM) caffeine. The tissues were washed with $4\,°C$ cold saline and immediately cryopreserved in liquid nitrogen. Finally, the tissue samples were then utilized for proteome and metabolome analysis, while the blood samples were employed to detect biochemical indices.
## Sample preparation for proteome analysis

Grinded frozen tissue samples were added to lysis buffer (8M urea, 1% protease inhibitor) and then lysed by sonication. The lysate was centrifuged at 12,000 $g$ for 10 min at 4°C to remove cell debris, and the supernatant was transferred to a new tube for protein quantification using BCA kit (Beyotime, Haimen, China). Equal amounts of protein were extracted from each sample and the volume was adjusted to the same volume with lysis buffer. The 20% Trichloroacetic acid (Sigma-Aldrich, Shanghai, China) was added, mixed by vortex, and precipitated at 4°C for 2 h. The precipitated sample was centrifuged at 4,500 $g$ for 5 min, the supernatant was discarded, and the precipitate was washed 2–3 times with pre-cooled acetone. Tetraethylammonium bromide (Sigma-Aldrich, Shanghai, China) with a concentration of 200 mM was added to the air-dried precipitate and then dispersed by ultrasonication. Trypsin was added at a ratio of 1:50 (protease : protein) for enzymatic hydrolysis overnight. After enzymatic hydrolysis, DL-Dithiothreitol (Sigma-Aldrich, Shanghai, China) was added to a final concentration of 5 mM, and reduced at 56 °C for 30 min. Iodoacetamide (Sigma-Aldrich, Shanghai, China) was added to a final concentration of 11 mM and incubated at room temperature for 15 min in the dark.

## Proteome LC-MS/MS analysis

The peptides were separated by ultra-high-performance liquid chromatography and injected into NSI source for ionization, and then analyzed by Orbitrap Exploris™ 480 mass spectrometer. The ion source voltage was set to 2.3 kV, the FAIMS compensation voltage (CV) was set to −45 V and −65 V, and the peptide precursor ions and their secondary fragments were detected and analyzed using high-resolution Orbitrap. The scanning range of the primary mass spectrometer was set to 400–1,200 m/z, and the scanning resolution was set to 60,000. The fixed starting point of the scanning range of the secondary mass spectrometer was 110 m/z. The secondary scanning resolution was set to 15,000, and the TurboTMT was set to off. The data acquisition mode used a cycle time-based data-dependent scanning program, that is, the peptide precursor ions were selected according to the order of signal intensity to enter the HCD collision cell within a cycle period of 1.0 s, and 27% fragmentation energy was used. Following the same procedure, the secondary mass spectrometry analysis was performed. In order to improve the effective utilization of mass spectrometry, automatic gain control was set to 100%, signal threshold was set to 5E4 ions/s, maximum injection time was set to auto, and dynamic exclusion time of tandem mass spectrometry scanning was set to 20 s to avoid precursor ions repeated scans.

Data of the secondary mass spectrometry was searched in Uniport Mus_musculuc database (17063 sequences) using proteome discoverer software (v2.4.1.15). An anti-library was added to calculate the false positive rate. Due to random matches, a common contamination library was added to the database to eliminate the effect of contaminating proteins in the identification results. Finally, the proteome quantitation data of each sample was obtained for subsequent analysis.

## Sample preparation for metabolome analysis

The frozen tissue samples were thawed on ice and 50 mg of the samples were weighed into a two mL EP tube, and 500 µL of −20 °C pre-cooled 70% methanol water internal

standard extraction solution was added. A small steel ball was added to the sample, and the sample was homogenized 4 times at 30 Hz for 30 s each time. After homogenization, the samples were shaken at 1,500 r/min for 5 min, and allowed to stand on ice for 15 min. After that, the samples were centrifuged at 12,000 r/min for 10 min at 4 °C. 200 μL of the supernatant was loaded into the lined pipe for subsequent analysis.

## Metabolome analysis

We selected five pathways involved in energy metabolism and performed widely targeted detection on 65 metabolites involved in them. Five energy metabolism pathways include glycolysis, citrate cycle, oxidative phosphorylation, amino acid metabolism and pentose phosphate pathway. The sample extracts from liver and muscle were analyzed using ultra-performance liquid chromatography-electrospray ionization-mass spectrometry (UPLC-ESI-MS/MS; UPLC, Shim-pack UFLC SHIMADZU CBM30A system, Shimadzu, Chengdu, China; MS, QTRAP® System; Sciex, Shanghai, China).

The chromatographic column for the liquid phase was an SeQuant ZIC-pHILIC, 5 μm (2.1 mm × 100 mm). In the mobile phase, phase A was 0.3% ammonia solution to which 10 mmol/L ammonium acetate was added and phase B was 90% acetonitrile water. The proportion of phase A and phase B in the elution gradient was 5:95 (V/V) at 0 min, 50:50 (V/V) at 9.5 min, 5:95 (V/V) at 11.1 min and 5:95 (V/V) at 14 min. The injection volume was 2 μL, with the flow rate set to 0.4 mL/min, and the column temperature set to 40 °C. The effluent was connected to a triple quadrupole-linear ion trap mass spectrometer (QTRAP) equipped with an ESI Turbo Ion-Spray interface. After the mass spectrum analysis, metabolite quantitation data for each sample were acquired through multiple reaction monitoring analysis. The peak area of each chromatographic peak represents the relative content of the corresponding substance. The chromatographic peak area was calculated and corrected by using MultiQuant software (v3.0.2) and the chromatographic peaks detected in different samples for each metabolite are calibrated to ensure qualitative and quantitative accuracy.

## Principal component analysis

The unsupervised principal component analysis (PCA) was performed on all detected proteins and metabolites using the statistical function prcomp in R (version 4.1.0; *R Core Team, 2021*). The 95% confidence ellipse was drawn according to the distribution of samples in different groups and colored by groups.

## Differential features selection

Statistical analysis was performed to select differential proteins and metabolites. Student's $t$-test was performed between the two groups. The $p$ values of differential proteins were corrected using the Benjamin-Hochberg method for multiple hypothesis testing. The corrected $p$ value ($q$ value) <0.05 and the absolute value of $\log_2$ FoldChange >0.5 were used as the threshold for significantly differential proteins. The significantly differential metabolites were filtered as the $p$ value <0.05 and the absolute value of $\log_2$ FoldChange(FC) >0.5. The variable importance in projection (VIP) value of differential metabolites presented in supplementary table was calculated through orthogonal partial least-squares

discrimination analysis (OPLS-DA) analysis using R package MetaboAnalystR (v1.0.1). According to these thresholds, the differential proteins and metabolites were presented by volcano plot using R package EnhancedVolcano (v1.14.0). The heat map was used to display the relative content of differential proteins and metabolites in each sample and the $z$ score was calculated by scaled as samples using R package pheatmap (v1.0.12). Hierarchical clustering was carried out on samples and features in heat map.

### Correlation analysis

Network analysis was performed between all differential proteins and differential metabolites with $p$ value <0.01 by calculating Spearman's rank correlation coefficient (Rho) using R package psych (v2.1.9). Benjamin-Hochberg correction was performed for the $p$ value of correlation analysis results. The correlations with adjusted $p$ value ($q$ value) <0.05 and rho >0.6 were retained and visualized by cytoscape software (v3.8.2). Linear correlations between the two metabolites were measured by calculating Pearson correlation coefficient (r) and visualized by using R package ggplot2 (v3.3.5).

### Biochemical indexes test

Lactate levels were quantified using Lactate Scout 4 (EKF Diagnostics Holdings plc, Cardiff, UK). Glycogen levels of liver was detected using glycogen assay kit ab65620 (ab65620, Abcam) as the glycogen assay protocol. Phosphoenolpyruvate (PEP) levels of liver was detected using PEP assay kit ab204713 (ab204713; Abcam). Pyruvate carboxylase (PC) levels of liver was detected using mouse PC ELISA kit (JYM1189Mo; ELISA Lab). The Optical Density (O.D.) was determined using a microplate reader (Thermo Fisher Scientific™ Varioskan™ LUX; Thermo Fisher Scientific, Waltham, MA, USA) in our lab. Only twelve samples in the discovery group were tested for glycogen, PEP and PC levels of liver due to insufficient tissue samples. Student's $t$-test was performed to acquire the statistical results between two groups.

## RESULTS

### Proteome of liver and gastrocnemius muscle

To investigate the proteome composition differences between groups, we initially conducted principal component analysis (PCA) to assess the clustering patterns of liver and gastrocnemius muscle samples (Fig. 1AB). Differential analysis revealed that there were 23 and 3 proteins exhibiting differential expression between CL *vs* FL and CM *vs* FM, respectively, with no interaction between them. Specifically, 13 proteins were found to be significantly up-regulated while 10 proteins were down-regulated in FL group (Fig. 1C, Table S1). Two proteins were up-regulated and one protein was down-regulated in the FM group compared to the CM group (Fig. 1D, Table S2). The relative abundance of differential proteins in each sample was displayed, and hierarchical clustering for samples was performed (Fig. 1EF). Based on 23 differential proteins, the CL and FL groups could be clearly clustered into two groups. Also, the CM and FM groups acquired a clear clustering result.

In the up-regulated proteins of the FL group, Rer1 exhibited a remarkable increase of 3.8-fold ($q < 0.01$) compared to CL. S100a9 and S100a8, both playing a prominent role in

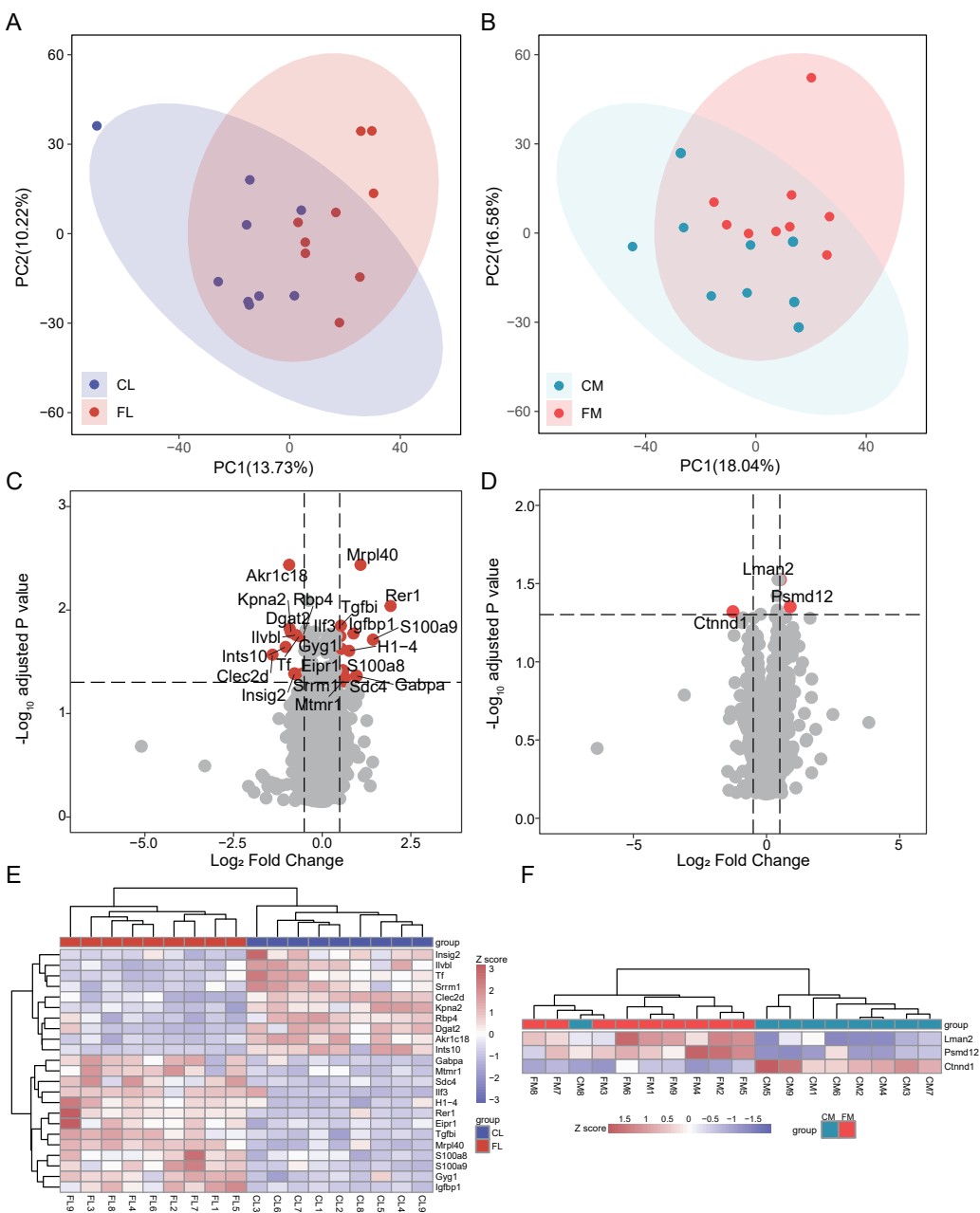

**Figure 1    Differential analysis in protein expression levels of liver and muscle between groups.** (A) PCA analysis of proteomic data of liver between control (CL) and caffeine (FL) groups. (B) PCA analysis of proteomic data of muscle between control (CM) and caffeine (FM) groups. (C) Volcano plot of significantly differential proteins of liver between CL and FL. (D) Volcano plot of significantly differential proteins of muscle between CM and FM. (E) Heat map of significantly differential proteins of liver between CL and FL. (F) Heat map of significantly differential proteins of muscle between CM and FM.

the regulation of inflammatory processes and immune response and were more expressed in the practitioners of physical exercise (*Pacheco et al., 2022*), were significantly elevated in FL group. The expression level of mitochondrial ribosomal protein L40 (Mrpl40) was twice that

of CL. GA-binding protein alpha chain (Gabpa), a sport-related gene identified in horses by genome-wide association study (*Schröder et al., 2012*), was significantly up-regulated in FL group. The expression of insulin-like growth factor-binding protein 1 (Igfbp1), known for its protective function against cardiovascular pathologies and strongly increased after swimming in mice and healthy people (*Hoene et al., 2010*; *Nishida et al., 2010*; *Rajwani et al., 2012*), was notably up-regulated in FL. Syndecan-4 (Sdc4), which was increased in human plasma after swimming exercise and in the livers of exercise trained mice compared with sedentary ones (*De Nardo et al., 2022*; *Lee et al., 2019b*), also showed a significant increase in FL. A significant reduction in tissue factor (Tf) induced by running (*Highton et al., 2019*) was also observed in FL group. Additionally, there was a notable increase in myotubularin-related protein 1 (Mtmr1), a lipid phosphatase with high specificity for phosphatidylinositol 3-phosphate, and glycogenin-1 (Gyg1), which initiates glycogen synthesis, within FL group. The expression of diacylglycerol O-acyltransferase 2 (Dgat2), responsible for triacylglycerol synthesis, and insulin-induced gene 2 protein (Insig2), involved in cholesterol synthesis and obesity, were down-regulated in FL. Retinol-binding protein 4 (Rbp4), which is associated with improved insulin sensitivity and exercise training (*Marschner et al., 2017*), was also decreased in FL group.

Differential analysis results showed that only two proteins were up-regulated in FM group, while one protein was down-regulated. These indicate that caffeine has minimal impact on the protein levels of gastrocnemius muscle after swimming. Notably, we found a significant up-regulation of Psmd12 in FM, which is involved in cell cycle progression, apoptosis and DNA damage repair.

## Energy metabolome of liver and gastrocnemius muscle

To investigate alterations of metabolites involved in energy metabolism, we conducted a comprehensive analysis of 65 metabolites associated with this process. Initially, PCA was performed to assess the metabolites detected in liver, revealing distinct separation between CL and FL (Fig. 2A). Similarly, PCA analysis of metabolites in gastrocnemius muscle indicated clear clustering between CM and FM (Fig. 2B). Subsequently, differential metabolite analyses were conducted for CL *vs* FL and CM *vs* FM separately, resulting in the identification of 19 and 22 significantly differentially expressed metabolites respectively (Fig. 2CD). Notably, nine of these differentially expressed metabolites were found to be common to both groups. The relative expression levels of these differential metabolites were displayed by a heat map and hierarchical clustering was applied to group samples accordingly (Fig. 2EF). The observed clustering pattern indicated that these specific metabolites effectively characterize each group.

The levels of amino acids, including asparagine, serine, tyrosine, cysteine, threonine and citrulline were significantly up-regulated in FL compared to CL group. Conversely, metabolites involved in glycolytic pathway such as glucose 6-phosphate, fructose 6-phosphate, pyruvic acid and lactate exhibited significant down-regulation in FL group indicating a decrease in glycolysis.

The differential analysis results between CM and FM showed a significant increase in dAMP, oxaloacetate, uracil, dUMP, phenyllactate and pyruvic acid levels in FM

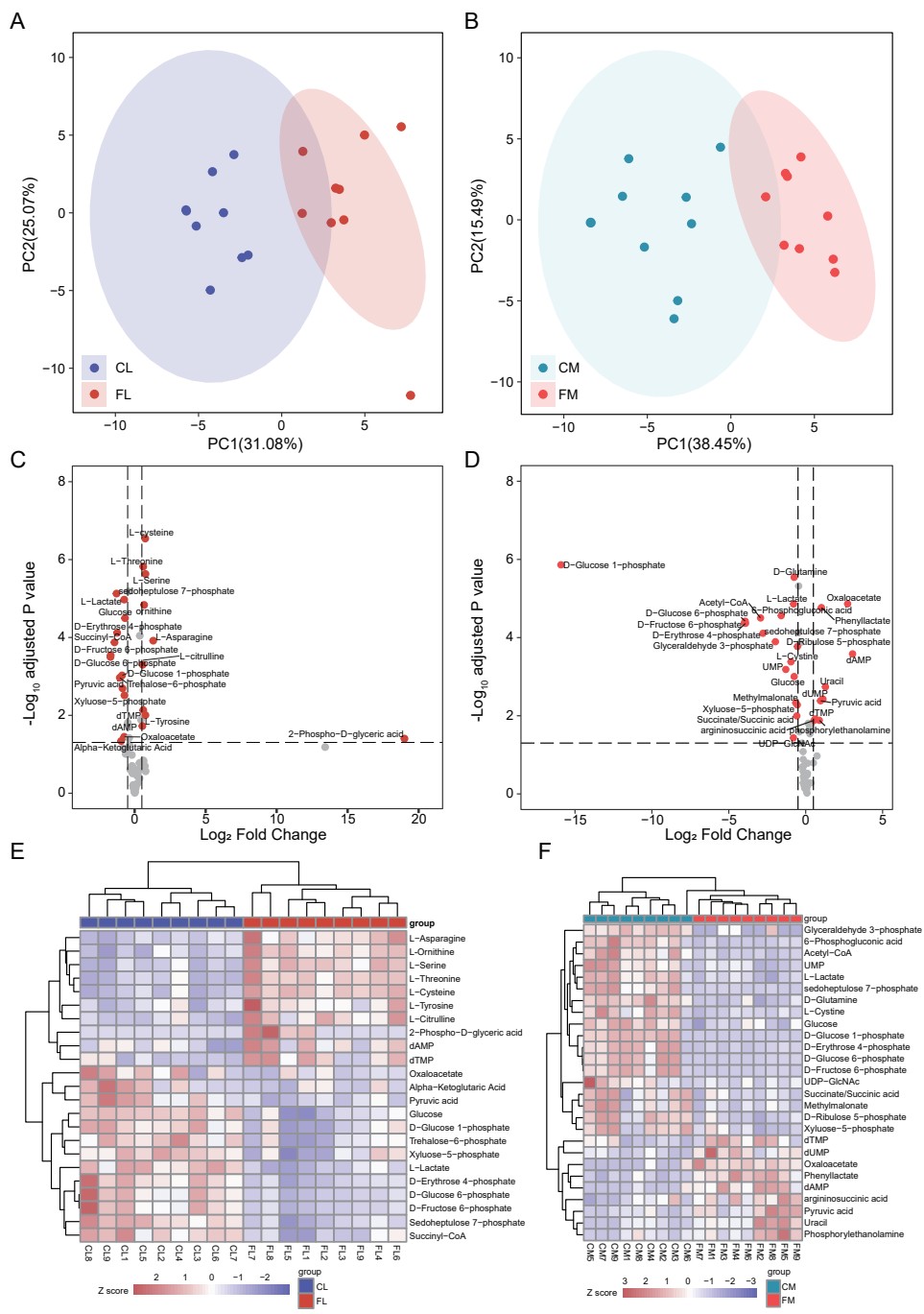

**Figure 2** **Differential analysis in metabolite abundance levels of liver and muscle between groups.** (A) PCA analysis of metabolic data of liver between control (CL) and caffeine (FL) groups. (B) PCA analysis of metabolic data of muscle between control (CM) and caffeine (FM) groups. (C) Volcano plot of significantly differential metabolites of liver between CL and FL. (D) Volcano plot of significantly differential metabolites of muscle between CM and FM. (E) Heat map of significantly differential metabolites of liver between CL and FL. (F) Heat map of significantly differential proteins of muscle between CM and FM.

group. Conversely, metabolites associated with the pentose phosphate pathway such

as glucose-6-phosphate, 6-phosphogluconic acid, ribulose-5-phosphate, xyluose-5-phosphate, sedoheptulose-7-phosphate and D-erythrose-4-phosphate were all found to be decreased in FM group. These findings suggest a down-regulation of the pentose phosphate pathway in FM group. Additionally, glycolysis pathway was also decreased in FM, due to the down-regulation of glucose, glucose-6-phosphate, fructose-6-phosphate and glyceraldehyde-3-phosphate.

## Correlation analysis

To investigate the associations between proteins and metabolites, we performed Spearman's rank correlation analysis separately for liver and muscle. Our results revealed that Sdc4, S100a9, S100a8, Gyg1 and Igfbp1 in liver were positively correlated with cysteine, threonine, ornithine, asparagine and serine (Fig. 3A). Furthermore, we observed a positive correlation between the reduction of glucose and proteins associated with alleviating obesity and type 2 diabetes, such as Insig2, Dgat2 and Rbp4. Additionally, Kpna2 exhibited a positive correlation with metabolites involved in glycolysis including D-glucose 6-phosphate, D-fructose 6-phosphate and pyruvic acid. Correlation analysis results for gastrocnemius muscle showed that Psmd12 was correlated with several significantly altered metabolites in FM group, such as lactate acetyl-CoA, glucose, D-glucose 6-phosphate and D-fructose 6-phosphate (Fig. 3B). Moreover, the up-regulated Lman2 showed a positive correlation with pyruvic acid and oxaloacetate.

To further investigate the associations between liver and muscle after caffeine administration, Pearson correlation analysis was performed for 38 metabolites and 26 proteins with significant differences between control and caffeine groups. Results showed that 38 metabolites were significantly associated with 25 proteins (Fig. 3C, $r > 0.3$, $q$ value <0.05, Table S5). Among these associations, Tgfbi and Gyg1 in liver, as well as Lman2 in muscle, emerged as three nodes with the highest degree within the network. Notably, glucose in muscle exhibited the maximum degree among all metabolite nodes. Additionally, strong positive correlations ($r > 0.9$) were observed between L-Asparagine and Gabpa, Ctnnd1, Gyg1 and Eipr1. Furthermore, decreased levels of glucose in muscle were positively correlated with Ilvbl but negatively correlated with Gyg1 in liver.

## Pathway analysis

To investigate changes in energy metabolism, we conducted pathways enrichment analysis on the differential metabolites. Our findings revealed significant alterations in glycolysis, citrate cycle and pentose phosphate pathways in the FL and FM groups. Glycolysis pathway annotated by KEGG with differential metabolites was presented (Fig. 4A). Notably, glucose, glucose 6-phosphate and fructose 6-phosphate were all significantly down-regulated in both FL and FM groups, indicating a substantial decrease in glycolysis within the liver and gastrocnemius muscle of caffeine-administered mice compared to the control group. Pyruvic acid and oxaloacetate levels were significantly decreased in FL, while increased in FM. As a result of down-regulated glycolysis, it is speculated that elevated pyruvic acid levels in gastrocnemius muscle may be produced by alternative pathways. Notably, serine and cysteine can be metabolized into pyruvic acid (Gray, Tompkins & Taylor, 2014), which

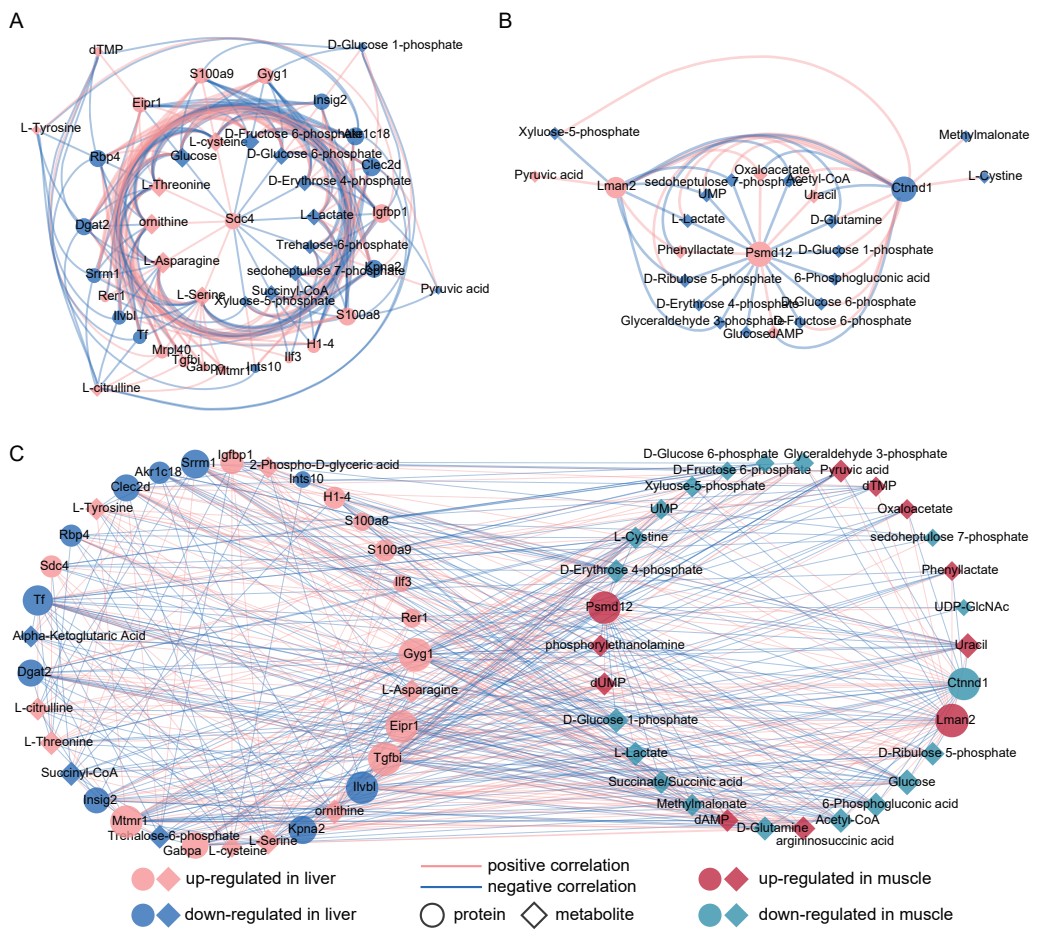

**Figure 3** (A–C) Correlation analysis between significantly differential proteins and metabolites in liver and muscle. (A) Spearman's rank correlation analysis of differential proteins and metabolites between control (CL) and caffeine (FL) groups. (B) Spearman's rank correlation analysis of differential proteins and metabolites between control (CM) and caffeine (FM) groups. (C) Pearson correlation analysis for all metabolites and proteins with significant difference between the control and caffeine groups ($r >$ 0.3, $q$ value <0.05).

is further catalyzed by pyruvate carboxylase to oxaloacetate. Our results indicate that serine and cysteine were significantly up-regulated in the FL group but not in the FM group. Furthermore, we compared the levels of pyruvate carboxylase (PC) between CL and FL groups. PC levels were measured in twelve samples from the discovery group, revealing a significant increase in FL compared to CL ($p = 0.045$, Fig. 4B). It was further validated in FL_V group compared to CL_V group ($p = 0.015$, Fig. 4B). These results suggest that the elevated levels of pyruvic acid and oxaloacetate in muscle may be synthesized through serine and cysteine in liver. Further analysis for amino acids revealed increased levels of tyrosine, threonine, ornithine and citrulline in the FL group. It is noteworthy that fumarate, succinyl-CoA and alpha-ketoglutaric acid involved in the citrate cycle could be synthesized through tyrosine, threonine, ornithine and citrulline metabolism, respectively. Interestingly, the levels of fumarate, argininosuccinic acid and malate were significantly

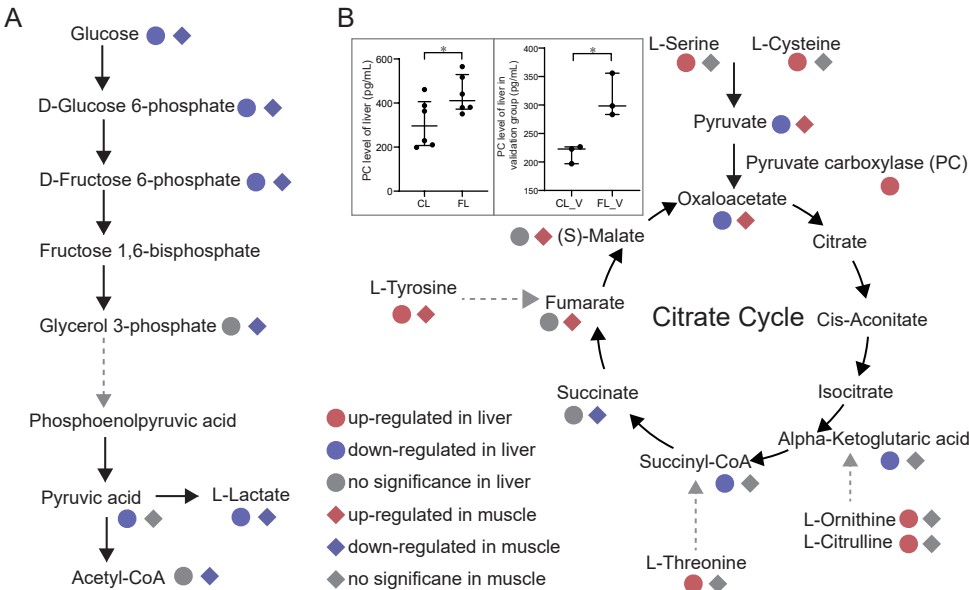

**Figure 4 Analysis of energy metabolic pathways.** (A) Glycolysis pathway and citrate cycle marked with differential metabolites between groups in liver and muscle. (B) Comparisons of pyruvate carboxylase level of liver between the control (CL) *vs* caffeine (FL) groups and control of validation (CL$_V$) *vs* caffeine of validation (FL$_V$) were displayed in the gray rectangle. *$p < 0.05$.

up-regulated in FM group. However, succinyl-CoA and alpha-ketoglutaric acid were significantly decreased in the FL group. These findings suggest a potential correlation between elevated amino acid levels in liver and increased metabolites associated with the citrate cycle in gastrocnemius muscle.

To further confirm our hypotheses, we performed Pearson correlation analysis between amino acids in liver and metabolites involved in the citrate cycle in gastrocnemius muscle. Our results showed significant positive correlations between tyrosine, serine, cysteine, ornithine, citrulline and threonine in liver and oxaloacetate in muscle ($p < $ 1e-4, $r \geq$ 0.8, Fig. 5A). Notably, these correlations were predominantly observed under the caffeine group rather than control group. Additionally, serine and cysteine in liver were significantly positively correlated with pyruvic acid in muscle ($p < 0.05$, $r \geq 0.5$, Fig. 5B). Consistent with our previous speculation, a significantly positive correlation was observed between pyruvic acid and oxaloacetate in muscle ($p < 0.01$, $r = 0.6$, Fig. 5B). Furthermore, correlation analysis results provide evidence suggesting that amino acids in liver may serve as potential precursors for the synthesis of pyruvic acid and oxaloacetate in muscle. However, imperfect correlations between pyruvic acid and serine or cysteine imply the existence of alternative pathways associated with increased levels of pyruvic acid. To investigate the influence of the Cori cycle, which facilitates lactate transport and its conversion to glucose in liver for subsequent utilization by muscle, we performed correlation analysis between lactate and pyruvic acid in muscle and liver. Results showed that lactate level was negatively correlated with pyruvic acid in muscle ($p = 0.0016$, $r = -0.69$, Fig. 5C). Additionally, we observed a

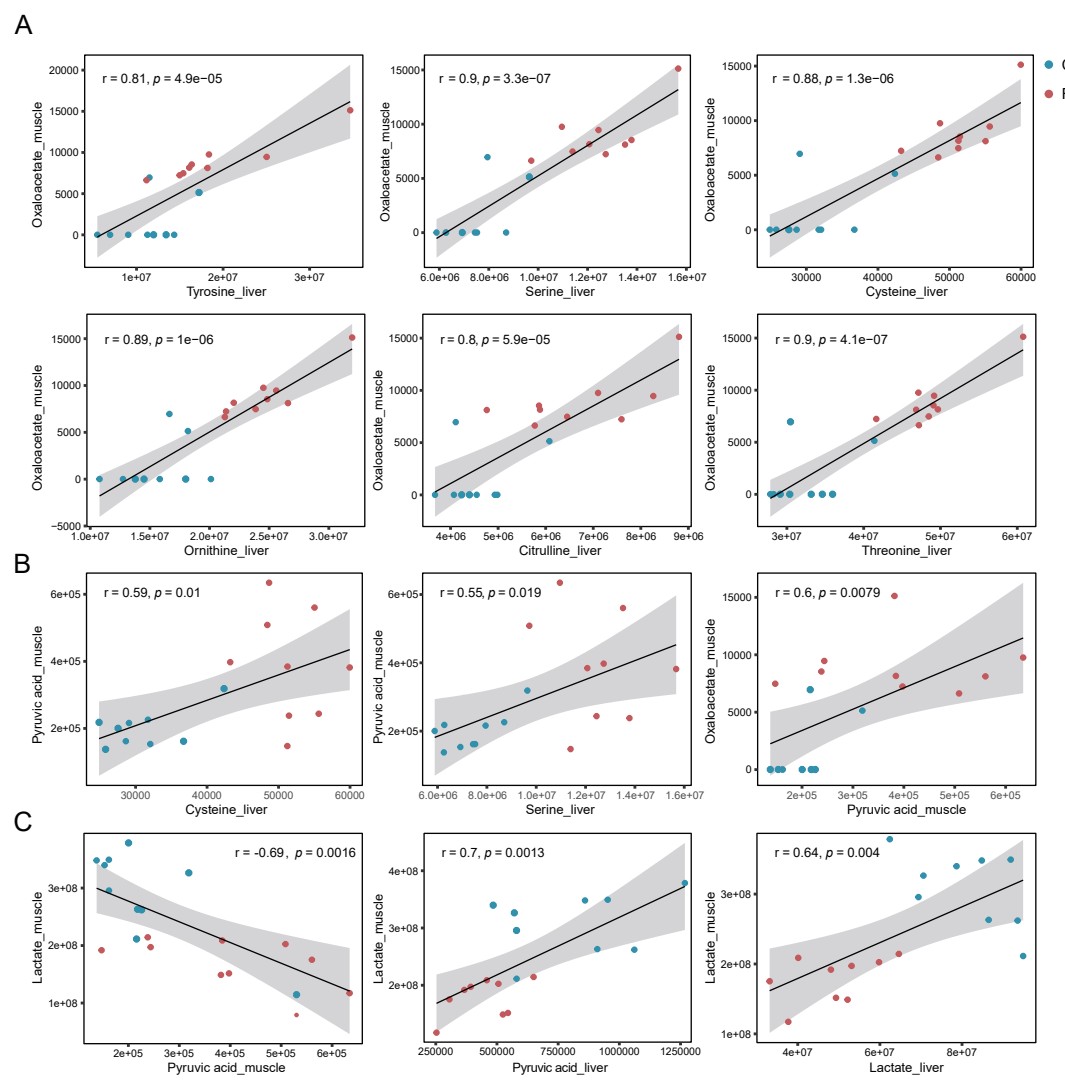

**Figure 5** (A–C) **Correlation analysis between metabolites of liver and muscle.** Pearson correlation analysis based on metabolome analysis results. The gray shade area represents the 95% confidence interval.

positive correlation between pyruvic acid levels in liver and lactate in muscle ($p = 0.0013$, $r = 0.7$, Fig. 5C). Furthermore, there was a significant positive association between the lactate levels in liver and lactate levels in muscle ($p = 0.004$, $r = 0.64$). Our results suggest that caffeine may contribute to elevated muscular pyruvate levels through its potential acceleration of the Cori cycle.

Additionally, we observed a significant decrease in the levels of metabolites generated through the pentose phosphate pathway, such as sedoheptulose-7-phosphate, D-erythrose-4-phosphate and xyluose-5-phosphate, in both the FL group and FM group. Furthermore, intermediates including D-ribulose-5-phosphate, 6-phosphogluconic acid, glyceraldehyde-3-phosphate were significantly down-regulated in FM group. This result suggests a

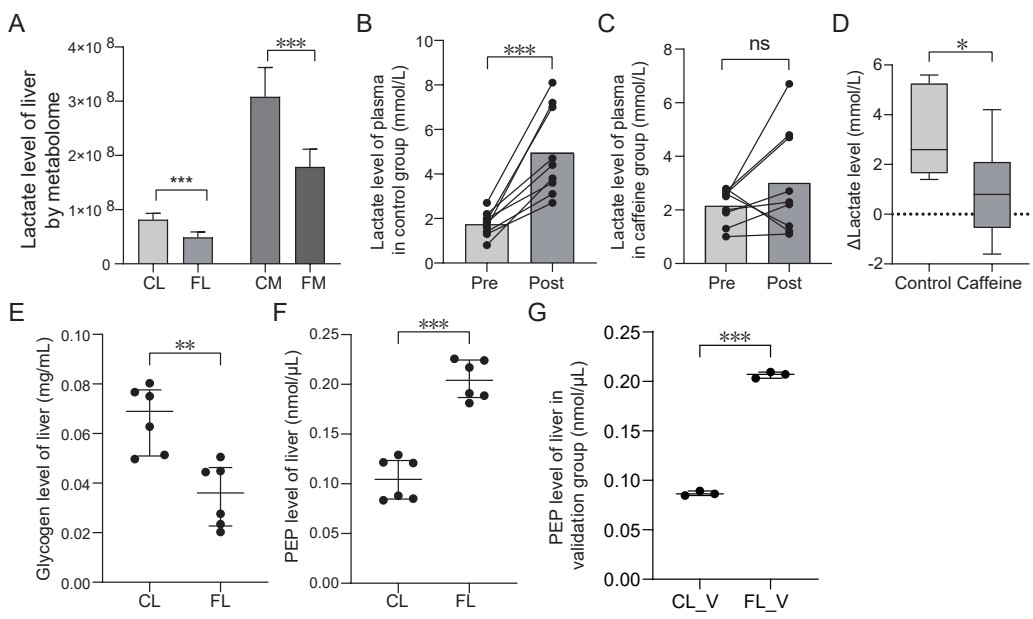

**Figure 6 Comparison of lactate, glycogen, phosphoenolpyruvate (PEP) levels between groups based on quantification test.** (A) Comparison of lactate level based on metabolome analysis. (B) Comparison of lactate level of plasma in control group between pre-swimming (pre) and post-swimming (post) based on quantification test. (C) Comparison of lactate level of plasma in caffeine group between pre-swimming (pre) and post-swimming (post) based on quantification test. (D) Comparison of difference of lactate level of plasma before and after swimming between control and caffeine groups. (E) Comparison of glycogen level of liver between control (CL) and caffeine (FL) groups based on quantification test. (F) and (G) Comparison of phosphoenolpyruvate level of liver between groups based on quantification test. $^*p < 0.05$, $^{**}p < 0.01$, $^{***}p < 0.001$, ns, no significance.

substantial down-regulation of the pentose phosphate pathway in both liver and muscle after caffeine administration.

## Caffeine promotes the utilization of lactate and glycogen

Energy metabolome analysis revealed that caffeine significantly decreased lactate levels in liver and muscle after mice swimming (Fig. 6A). We speculated that caffeine accelerates the consumption and the reduction of lactate. To further substantiate the role of caffeine in enhancing lactic acid utilization, we quantified the plasma lactate levels before and after swimming in control and caffeine groups. Remarkably, post-swimming plasma lactate levels exhibited a significant increase in control group ($p < 0.001$, Fig. 6B). However, in caffeine group, there were no significant changes observed in plasma lactate levels after swimming ($p = 0.19$, Fig. 6C). The difference in lactate levels before and after swimming ($\Delta$ Lactate) was found to be significantly different between control and caffeine groups ($p = 0.01$, Fig. 6D). Our results validate that caffeine enhances the utilization of post-exercise lactate, thereby maintaining its levels equivalent to pre-exercise conditions.

To further investigate the impact of caffeine on liver glycogen levels after swimming, we compared the glycogen levels between control and caffeine groups. Glycogen levels in the discovery group showed a significant decrease in the FL group compared to the CL group

($p < 0.01$, Fig. 6E). These results indicate that caffeine enhances glycogen utilization in liver after swimming. Previous studies have shown that exercise training increases hepatic gluconeogenesis after swimming to resist to exercise-induced hypoglycemia (*Bergman et al., 2000*; *Donovan & Sumida, 1997*). Oxaloacetate, an intermediate metabolite in gluconeogenic pathway, can be converted into phosphoenolpyruvate (PEP) and eventually glucose. Therefore, we examined PEP levels in liver between the CL and FL groups. PEP levels in the discovery group showed a significant increase in FL group ($p < 0.001$, Fig. 6F). This result was further validated by analyzing another independent sample set ($p < 0.001$, Fig. 6G). These results revealed that exercise with caffeine intake significantly improved exercise-induced hypoglycemia, but cannot explain its effect on liver gluconeogenesis (*Gilglioni et al., 2016*). Our results indicate that caffeine administration before exercise promotes increased conversion from oxaloacetate to phosphoenolpyruvate, potentially accelerating liver gluconeogenesis.

## DISCUSSION

The objective of this study is to investigate the impact of caffeine supplementation before exercise on the proteome and energy metabolism after mice swimming. There is convincing evidence that caffeine supplementation can be conducive to muscular endurance and power in resistance exercise (*Grgic, 2021*). The previous study also demonstrated that caffeine supplementation enhances anaerobic performance in the a 30-seconds all-out Wingate test (*San Juan et al., 2019*). Recent study indicated that a single session of resistance exercise training effectively enhances glycolysis, thereby facilitating the rapid provision of ATP for muscle contractions (*Koh et al., 2022*). Metabolites involved in the glycolysis pathway exhibited significant down-regulations in liver and muscle after caffeine administration in our study. Therefore, the ergogenic effects of caffeine on exercise performance may be partially attributed to its ability to expedite metabolite utilization in glycolysis during exercise.

Our study showed some proteins like S100a8, S100a9, Gabpa, Igfbp1 and Sdc4 associated with excellent exercise performance highly expressed in liver of mice administered with caffeine. S100a8 and S100a9 play prominent roles in the regulation of inflammatory processes and immune response (*Boyd et al., 2008*), and both were more expressed in the practitioners of physical exercise (*Pacheco et al., 2022*). Gabpa plays an essential, nonredundant role in mitochondrial biogenesis (*Yang et al., 2014*), and was identified as a sport-related gene (*Schröder et al., 2012*). Igfbp1 is an important modulator of the insulin-like growth factors that may have both positive and negative effects on the ability of insulin-like growth factors to stimulate cell growth (*Lee et al., 1994*). Sdc4 is a cell surface proteoglycan which regulates exosome biogenesis, and it has been proven that the decreased expression of Sdc4 leads to reduced skeletal muscle performance in mice (*Sztretye et al., 2023*). These proteins potentially contribute to the advantageous impact of caffeine on exercise performance, which might be a promising avenue for further investigation. Currently, limited research has been conducted on the impact of caffeine on the proteome after swimming. To our knowledge, our study firstly identified the up-regulation of
exercise-related proteins induced by caffeine after swimming, thereby improving exercise performance possibly.

Previous study has demonstrated that pre-exercise oral lactate intake enhances the activity of mitochondrial cytochrome c oxidase in the soleus muscle of mouse (*Takahashi et al., 2020*). Furthermore, lactate administration increases mRNA expression of protein synthesis factors in skeletal muscle, suggesting its potential role in promoting muscle synthesis (*Kyun et al., 2020*). Recent study reveals that acute lactate administration augments carbohydrate metabolism after swimming exercise in mice (*Kyun et al., 2020*). These findings highlight the significant involvement of lactate in energy metabolism. Moreover, several studies have confirmed lactate's ability to serve as an energy source for tissues and organs during physical exercise (*Brooks, 2018*; *Gladden, 2004*; *Philp, Macdonald & Watt, 2005*). Our study demonstrates that caffeine significantly reduces post-swimming lactate levels, suggesting enhanced utilization of lactate after caffeine consumption. Additionally, decreased post-swimming lactate levels induced by caffeine administration contribute to an elevated lactate threshold, thereby benefiting exercise performance possibly.

In caffeine group, we observed a positive correlation between the elevated levels of serine and cysteine in liver and pyruvate and oxaloacetate in muscle. Additionally, there was also a positive correlation between pyruvate and oxaloacetate levels in muscle. However, these correlations were not observed in the control group. Since oxaloacetate can be synthesized from serine and cysteine *via* pyruvate, and we also demonstrated that pyruvate carboxylase was elevated in the caffeine group, it is possible that caffeine accelerates the synthesis of pyruvate and oxaloacetate in muscle from amino acids in liver.

In addition, the administration of caffeine in mice resulted in decreased or increased levels of certain proteins, such as Insig2, Rbp4, and Igfbp1, which have been associated with improved obesity, type 2 diabetes mellitus (T2DM), and cardiovascular pathologies. Numerous studies have indicated that coffee consumption is linked to a reduced risk of diseases, such as T2DM (*Nieber, 2017*), obesity (*Lee et al., 2019a*) and cardiovascular disease (*Chieng & Kistler, 2021*). We speculate that the beneficial effects of coffee on these diseases may be attributed to alterations in gene expression caused by caffeine. Furthermore, we observed a significant up-regulation of Gabpa in the caffeine group. Notably, Gabpa has been demonstrated as a tumor suppressor in bladder cancer (*Guo et al., 2020*). Although the mechanism remains unclear, some studies suggest that tea containing caffeine can act as a preventive or therapeutic factor against bladder cancer. (*Conde et al., 2015*; *Yang et al., 2007*). Therefore, the protective effect of tea on bladder cancer may also involve an up-regulation of Gabpa expression induced by caffeine. Additionally, several proteins that drove diseases such as RER1 enhancing carcinogenesis and stemness of pancreatic cancer (*Chen et al., 2019*) and Igfbp1 confirmed as stroke risk marker were also increased in liver after caffeine administration. In summary, our results identified specific disease-related proteins regulated by caffeine, and the relationships between caffeine and these diseases need further exploration.

Although caffeine as a highly popular ergogenic aid and its beneficial effects on exercise performance has been admitted, further research is essential to investigate its underlying

mechanisms for future utilization. In regard to limitations, our study did not include a non-exercise control group since all animals underwent exercise training. Therefore, the relationship between changes in protein expression and energy metabolism induced by caffeine in non-exercise state cannot be explored. In addition, based on our experimental design, it is challenging to elucidate the process behind liver amino acids promoting muscle pyruvate and oxaloacetate synthesis. Nevertheless, our study provides evidence supporting the impact of caffeine on energy metabolism and protein expressions after swimming, and extends our knowledge in mechanism behind the effects of caffeine on exercise. In future, we will focus on identifying specific targets of caffeine action and exploring the process involved in reducing lactate levels post-exercise.

## CONCLUSION

Our study highlights the alterations of proteome and energy metabolome of liver and muscle induced by caffeine after swimming. Caffeine significantly upregulates the expression of exercise-related proteins, and decreased lactate levels after exercise. What's more, caffeine potentially enhances the utilization of glycolytic intermediates and facilitates the conversion of amino acids in liver to pyruvate and oxaloacetate in muscle after exercise.

### Funding
This study was supported by the Independent Research Project of Medical Engineering Laboratory of Chinese PLA General Hospital (grant number 2022SYSZZKY22) and National Foundation Strengthening Project (grant number 2020-JCJQ-ZD-260-12). The funders had no role in study design, data collection and analysis, decision to publish, or preparation of the manuscript.

### Grant Disclosures
The following grant information was disclosed by the authors:
Independent Research Project of Medical Engineering Laboratory of Chinese PLA General Hospital: 2022SYSZZKY22.
National Foundation Strengthening Project: 2020-JCJQ-ZD-260-12.

### Competing Interests
The authors declare that they have no competing interests.

### Author Contributions
- Yang Han conceived and designed the experiments, performed the experiments, analyzed the data, prepared figures and/or tables, authored or reviewed drafts of the article, and approved the final draft.
- Qian Jia conceived and designed the experiments, performed the experiments, prepared figures and/or tables, and approved the final draft.
- Yu Tian conceived and designed the experiments, performed the experiments, prepared figures and/or tables, and approved the final draft.
- Yan Yan performed the experiments, authored or reviewed drafts of the article, and approved the final draft.
- Kunlun He conceived and designed the experiments, authored or reviewed drafts of the article, and approved the final draft.
- Xiaojing Zhao conceived and designed the experiments, authored or reviewed drafts of the article, and approved the final draft.

### Animal Ethics

The following information was supplied relating to ethical approvals (i.e., approving body and any reference numbers):

This study was approved by the Animal Care and Use Committee of Chines PLA General Hospital (SQ2020030).

### Data availability

The dataset produced and/or analyzed in this study are available in the Supplementary Files.

### Supplemental Information

Supplemental information for this article can be found online at http://dx.doi.org/10.7717/peerj.16677#supplemental-information.

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
