# Peer review of "Multi-omics reveals changed energy metabolism of liver and muscle by caffeine after mice swimming"

_PeerJ, doi:10.7717/peerj.16677_

## Round 0.1 · original submission · Major Revisions

Please address concerns of all reviewers and amend manuscript accordingly.

**Language Note:** The review process has identified that the English language must be improved. PeerJ can provide language editing services - please contact us at copyediting@peerj.com for pricing (be sure to provide your manuscript number and title). Alternatively, you should make your own arrangements to improve the language quality and provide details in your response letter. – PeerJ Staff

Reviewer 1 ·

Basic reporting

With use of omics approaches, the authors investigated the effects of caffeine administration on substrate metabolism and protein content in the liver and skeletal muscle. However, as the authors repetitively introduce previous studies regarding caffeine supplementation, I do not understand the novelty and originality of the current report. The authors should clearly and concisely mention them. Additionally, the Discussion is composed based not on the current findings, but on their assumptions. At the current status, this article does not deserve publication.

Experimental design

L85-86
The method of administration should be clearly noted. What was the concentration? Was it intraperitoneally? In this case, how this study is relevant to practical settings, given that caffeine is generally consumed orally.

In regard with this, how the authors decided the caffeine volume.

L94-98
Although the authors used anesthesia for humane reasons, the effects of anesthesia should not be ignored.

L98-99
Centrifugation should be expressed as gravity, not roles/min, because the gravity can change depending on apparatus.

L99-103
Assuming that the blood was left for 40 min without adding anticoagulant, the blood supernatant is serum.

Additionally, during swimming exercise, upper limb muscles are recruited compared to lower limb muscles. There should be reasons why the authors used gastrocnemius muscles.

Validity of the findings

Nothing to mention.

Additional comments

L54-56
Although lactate can be converted to glucose in the liver, the major fate of lactate is oxidation in the working muscle.

L383-388
English can lead to misunderstandings. I wonder if this is about protein content as a result of long-term training or metabolite concentrations after single session of exercise.

L386-389
The authors say “the glycolysis pathway was significantly down-regulated in liver and muscle after caffeine administration” in the first sentence. However, the second sentence says “the beneficial effects of caffeine on exercise performance may be partly due to accelerated glycolysis during anaerobic exercise, which provides more energy for exercise.” The authors should modify this inconsistency.

L389-391
Metabolite concentrations in the specific pathway and cycle depend on the balance between metabolite flux into and out of them, and production and disposal. Without providing data, the authors should delete such an assumption.

L392-398
The authors listed proteins found to be differentially expressed. However, how those proteins are associated with exercise performance. Physiological and biochemical functions of those proteins should be mentioned in the article.

L410-412
Lactate is not a factor that induces muscle soreness and fatigue.

How do the authors interpret from correlation studies. Although the authors shows many correlation between two metabolites, there is no discussion regarding these observations, serving no purpose.

Overall
The authors performed omics analysis after exercise. I think pre-exercise protein expression is more relevant to subsequent exercise performance.

Lastly, it is unclear that the observed changes are induced by caffeine, exercise, or combined effects of them. To address this issue, the authors should include groups without exercise.

Reviewer 2 ·

Basic reporting

In this study, Han et al. employed mice as experimental models to investigate the impact of caffeine on the metabolic processes occurring in the liver and muscle following swimming. The research involved the utilization of mass spectrometry, metabolome analysis, and bioinformatics techniques. Through their analysis, the authors identified distinct sets of proteins that exhibited differential expression between the control group and the group treated with caffeine after swimming. Furthermore, by establishing connections between protein expression levels, metabolite levels, and metabolic pathways, the authors revealed that caffeine enhances the utilization of lactate and glycogen. In summary, this manuscript is professionally written and presents its findings in an easily comprehensible manner.

Experimental design

Regarding the experimental setup, it's worth noting that the mouse-forced swim test is a commonly employed method in the examination of depression and the assessment of the impact of antidepressants (Cryan et al., 2000; Lucki, 1997). It is necessary to provide a rationale for the choice of the forced swim test over voluntary wheel running when investigating the influence of caffeine on exercise. Additionally, it is important to address how the researchers have taken steps to separate the effects of exercise from those of stress when utilizing the swim test. Another concern is the absence of a control group receiving caffeine without engaging in swimming (sedentary), so it is unable to confirm whether the observed effects are a result of both caffeine and swimming or solely attributable to caffeine itself.

Validity of the findings

Unless I have misunderstood, Figure 4A appears to depict a downregulation of pyruvic acid in the liver with seemingly no effect on muscle, while Figure 4B shows an upregulation of pyruvate in muscle. To enhance clarity, I recommend that the authors consider using symbols (such as grey-color-filled circles and diamond shapes) to indicate metabolites for which no statistically significant differences between groups were observed. Regarding lines 302-304, I find it challenging to understand the rationale behind the relationship between oxygen usage and the differences in acetyl-CoA levels between CM and FM. For line 313-315, and 320-322, I think these claims are premature, given that this study primarily concentrates on the liver and muscle tissues. As for Figure 6, I suggest that the authors include a few non-correlated examples to provide a more comprehensive view of their findings and a comprehensive list of the r and p values resulting from the Pearson correlation analysis for all metabolites and amino acids with significantly different production levels.

Reviewer 3 ·

Basic reporting

The english is poor in some paragraphs. I suggest rewriting some parts of the text.

- The introduction is ok, but I missed some citations and reference, mainly in the discussion.

- Structure conforms to PeerJ standards, discipline norm, or improved for clarity.

- Figures are relevant, high quality, well labelled and described and raw data supplied.

- Self-contained with relevant results to hypotheses.

Experimental design

- Original primary research within Scope of the journal.

- Research question well defined, relevant and meaningful. It is stated how the research fills an identified knowledge gap.

- Rigorous investigation performed to a high technical and ethical standard.

- In methods, it is not clear if caffeine was given before or after swimming. And what route caffeine was given, injection or gavage?

Validity of the findings

- Impact and novelty assessed. However, I think is important to highlight and reinforce the importance of the novelty of the study, because there are scarce studies about molecular mechanisms and caffeine treatment after exercise.

- All underlying data have been provided; they are robust, statistically sound, & controlled.

- The conclusions was poor writen, rewrite the part of mechanisms.

Additional comments

- In this study, mice swimming experiment as the exercise model was conducted.
Instead I suggest: in this study, an exercise swimming model was conducted.

- In abstract is not clear if the caffeine was administered before or after exercise, important to highlight this in detail.

- The lactate levels of liver and muscle were significantly down-regulated by
caffeine after swimming, which partially indicated the benefitial effect of caffeine on exercise in mechanism. The conclusion is confusing, rewrite.

- The english is poor. I suggest rewriting various paragraphs, to improve english.

- Caffeine is very popular because its ergogenic or other cause? Citations?
- Mechanically or mechanistically?

- It was not clear if the caffeine was given before or after swimming. And in which Route was caffeine were given. Gavage, the injections. What are the directions and future studies in the field.

- Include more citations and references.

---

## Round 0.2 · accepted · Accept

All issues pointed out by the reviewers were addressed and the amended manuscript is acceptable now.

Reviewer 1 ·

Basic reporting

The authors improved their writings. Objectives and physiological rationales are now much clearer than before. The manuscript can be acceptable. Thank you.

Experimental design

no comment

Validity of the findings

no comment

Reviewer 2 ·

Basic reporting

No comment

Experimental design

The authors have answered all my questions.

Validity of the findings

No comment

Additional comments

Although the revised manuscript presents the data better, I think the writing and formatting still need more efforts.